# Effect of Selected Cooling and Deformation Parameters on the Structure and Properties of AISI 4340 Steel

**DOI:** 10.3390/ma13235585

**Published:** 2020-12-07

**Authors:** Maros Eckert, Michal Krbata, Igor Barenyi, Jozef Majerik, Andrej Dubec, Michal Bokes

**Affiliations:** 1Faculty of Special Technology, Alexander Dubcek University of Trencin, 911 06 Trencin, Slovakia; michal.krbata@tnuni.sk (M.K.); igor.barenyi@tnuni.sk (I.B.); jozef.majerik@tnuni.sk (J.M.); 2Faculty of Industrial Technologies, Alexander Dubcek University of Trenčín, 020 01 Puchov, Slovakia; andrej.dubec@tnuni.sk; 3Faculty of Military Technology, University of Defence in Brno, 662 10 Brno, Czech Republic; michal.bokes@unob.cz

**Keywords:** dilatometry, CCT diagram, true stress, true strain

## Abstract

The paper is focused on investigation of the high-strength AISI 4340 steel at various temperature and deformation conditions. The article is divided into two specific analyses. The first is to examine the dilatation behavior of the steel at eight different cooling rates, namely, 100, 10, 5, 1, 0.5, 0.1, 0.05 and 0.01 °C·s^−1^. The mapping of the phase transformations due to varying cooling rates from the austenitizing temperature of 850 °C allows the construction of the CCT diagram for a given high-strength steel. These dilatation curves were also compared with the metallography of the selected samples for the proper construction of the CCT diagram. A further analysis of the high temperature deformation of high strength steel AISI 4340 was performed in the range of temperature 900–1200 °C, and the strain rate was in the range from 0.001 to 10 s^−1^ with maximum value of the true strain 0.9. Changes in the microstructure were observed using light optical microscopy (LOM). The effect of hot deformation temperature on true stress, peak stress and true strain was investigated. The hardness of all deformed samples, depending on the temperature, the deformation rate and the peak stress σ_p_ overall together related with hardness, has also been evaluated.

## 1. Introduction

Deformation and dilatometric analysis are nowadays a widespread method for evaluating changes in the microstructure in steels, and at the same time they are associated with changes in the resulting mechanical properties [1]. Phase changes are the main goal of these measurements in the case of dilatometric examination. The influence of the heating rate on the austenitic temperature and its subsequent holding time plays a very important role in the dilatometry process [2]. Most experiments focus more on lower carbon steels [3,4]. Low or medium alloy steels, which are classified as ferritic steels, are also investigated [5,6]. The investigated steel AISI 4340 is classified as a medium alloy steel, the main alloys are Ni, Cr, Mo, V. Due to its mechanical properties and high strength, the investigated steel belongs to special high-strength steels, which are used for manufacturing of military equipment. Steel is used mainly on tank barrels and other highly loaded parts in this technique due to good material fatigue resistance.

Experimental steel AISI 4340 shows the presence of several types of carbides in the material microstructure. These carbides are mainly based on Fe, Ni, Cr, Mo and V. The occurrence of mainly these types of carbides M_2_C, M_7_C_3_, M_23_C_6_ and M_3_C is mentioned [7,8]. These hard carbide particles contribute to grain refinement by retarding the recrystallization of austenite and causing the strengthening by this mechanism [9]. Alloying element presence also noticeably affects austenite transformation, mainly through the shift of start and finish temperatures of phase transformations [10].

The results obtained by the dilatometer are used to construct various types of diagrams that are closely related to the heat treatment of steels. One of these diagrams is the Continuous Cooling Transformation (CCT) diagram. These focus on the anisotropic decay of austenite and forming new structures. During the measurement, the expansion curve deviation from its linear direction during cooling or heating is monitored. Based on this deviation, it is possible to determine the initial temperatures *Ac*_1_ and *Ac*_3_.

The CCT diagram gives information about the hardening response of steels and the nature of austenite transformation at varying degrees of cooling. This diagram has great practical importance to some heat treatment process such as austempering and isothermal annealing. The chemical composition of the investigated steels plays the most important role in the given process and influences the formation of the final microstructure. Likewise, the cooling rate of the sample is a very important factor in this process, examined by Lin et al. [11].

Phase transformations caused by temperature change can also be noticeably affected by application of proper deformation resulting to significant change of mechanical properties [12]. Application of the deformation brings effect at wide range of temperatures. If the deformation is applied at high temperatures with recrystallization effect, austenitic grain refinement occurs due to the repeated recrystallization what leads to improvement of mechanical properties. More noticeable effect to microstructure occurs at lower temperatures without recrystallization effect. The austenitic grains remain deformed and the deformation strips are formed inside them. Consequently, new phase (martensite) nucleates not only as a standard at grain boundaries but also inside the grain at the deformation strips. Nucleation within the deformed austenitic grains is one of the most important aspects of thermo-mechanical processing [13,14,15].

The material acts against every deformation, and this reaction is called deformation resistance. The deformation resistance is related to the temperature, deformation rate and the value of the deformation. Therefore, stress affecting the material must be higher than deformation resistance in order to achieve plastic deformation and consequently required shape change of the material. The forging of semi- products at higher temperatures is also one of the fundamental steps in the production of advanced high-strength steel. The combination of phase transformation and the forging (deformation) allow significant increase of the steel strength properties. The investigations using quenching and deformation dilatometer were carried out with controlling different deformation parameters, i.e., the degree of deformation (by how much the sample is to be deformed), the deformation rate and the temperature at which the deformation occurs [16,17].

The first goal of this paper is to investigate phase transformations of AISI 4340 high strength steel depending on the different cooling conditions. Subsequently, the creation of a CCT diagram for a given steel based on the achieved experimental data. The second goal is to investigate the deformation behavior of the steel at high temperatures and then to create corresponding True Stress–True Strain diagrams and investigate the relation between deformation parameters and deformation resistance. The contribution of the work is a complex evaluation of the investigated material AISI4340 from the point of view of thermo-mechanical processing and determination of the influence of selected parameters on the change of the experimental material structure in the monitored processes. It is also beneficial to create a corrected constitutive model in terms of friction during compression.

## 2. Materials and Methods

### 2.1. Experimental Material

The experimental steel that was used for the measurements is referred to as AISI 4340. Steel (Konstrukta Defence; Trencin, Slovakia) is used for components in special technology, withstands high fatigue loads and is characterized by high strength while maintaining sufficient toughness.

The chemical composition of the delivered steel was verified (Table 1). This analysis was performed on a special Q4 TASMAN chemical analyzer. Verification of the chemical composition was compared with the ISO 4967 standard. Table 2 also shows the basic mechanical properties of the examined steel [18].

Microstructure of investigated AISI 4340 steel (Figure 1a) was analyzed with using of light microscope Neophot 32 (ZEISS, Oberkochen, Germany) as well as SEM microscope Tescan Vega (Tescan, Brno, Czech Republic) (Figure 1b). The samples were etched by Nital (3% solution of nitric acid in ethanol). The microstructure of the supplied material is made of tempered martensite, bainite and fine secondary carbides, which are spherical in shape. The final structure is coarse-grained and relatively heterogeneous. The microstructure evaluated corresponds to the state after quenching and tempering of thick-walled Cr-Ni-Mo-V steels.

Two different martensite morphologies occurred in steel in the delivered state. The first form was represented by lath martensite, and the second was the occurrence of plate martensite. Lattice martensite occurs predominantly in steels that have a lower carbon content and medium content of additive alloying elements. In contrast to steels with a medium carbon content, martensite morphology is formed with both types. Small dimension of the lath martensite structure brings significant effect on the strengthening mechanism.

### 2.2. Continuos Cooling Transformation

The process in which homogeneous austenite decays into other structures during continuous cooling is called anisothermal decay of austenite. The resulting CCT diagrams express a graphical representation of this process. The cooling rate, which affects the formation of the final structure, has an analogous effect on the resulting decomposition of austenite [19,20].

All dilatometers utilize measurement of dimensions (length and/or diameter) and temperatures during heating and cooling. Each material consists of millions of atoms that are interconnected, and due to the increase in temperature, these atoms begin to move (oscillate). The total volume of material begins to change, which is associated with a change in the length of the material. The thermal expansion of all crystalline substances is directly related to the action of internal atomic forces in the crystal lattice [21].

Depending on the temperature, the resulting phase change reflects itself in a linear dilatation curve as a deviation of the direction of this curve, and its length changes on the experimental material. The temperature *Ac*_1_ examines the start of austenite formation when the material is heated due to the deviation of the dilatation curve from its linear direction. The final transformation of austenite formation occurs at the temperature *Ac*_3_, where again the dilatation heating curve begins to have its linear character.

Phase transformation studies were carried out using DIL805A/D dilatometer (TA Instruments, Milford, MA, USA), and the critical transformation points (temperatures) were determined according to [22,23,24]. The heating rate was 1 °C∙s^−1^ with holding time 30 min. at an austenitizing temperature of 850 °C. The cooling rate ranged from 100 °C∙s^−1^ to 0.01 °C∙s^−1^. Cooling regimens used for construction the CCT diagram of AISI 4340 steel are shown in Figure 2.

Samples were made in form of cylinder with diameter of 4 mm and length of 10 mm. Each experiment was repeated 4 times, and averages were calculated to obtain relevant values. The CCT diagram also shows the resulting hardness values, which were measured using Instron Wolpert (Instron, Norwood, MA, USA) hardness tester. Vickers method HV5 was used with load 50 N. Hardness was measured 5 times on each sample.

### 2.3. Investigation of Rheology

Deformation dilatometer (TA Instruments, Milford, MA, USA) used in this study utilizes compression of the samples in axisymmetric test. The results of the experiments are presented in form of true stress–true strain curves, Figure 3. It is a relation between the true stress and corresponding true strain, also called flow stress curves. Elastic deformation has a negligible value and therefore is omitted. There are four characteristic points in the curve. The point [0; *σ*_0_] is onset of plastic deformation. Stress *σ*_0_ could be considered as yield point in a compression. The point [*φ_p_;*
*σ_p_*] is the maximal value of stress, marked as *σ_p_*. Several authors describe the value as a Peak Stress [25,26].

Consequently, the strain value corresponding with the peak stress is described as peak strain. The coordinates [*φ_i_;*
*σ_i_*] define the inflectional point of the curve. The point [*φ_ss_;*
*σ_ss_*] is an equilibrium point from which strengthening and softening mechanisms are in balance. Therefore, the meaning of lower index “ss” is steady state.

Defined input parameters for compression test using of DIL 805A/D are temperature (*t),* deformation (φ) and deformation rate ($\dot{\varphi }$). The experiments were performed at temperatures of 900, 1000, 1100 and 1200 °C; strain rate from 0.001 to 10 s^−1^ and maximum strain 0.9. At the end of experiment, all samples were cooled at a constant cooling rate 20 °C/s. Samples were made in form of cylinder with diameter of 5 mm and length of 10 mm. Moreover, in these experiments, average values were obtained from 4 identical measurements to obtain relevant values. Due to the friction of the contact surfaces between the sample and the compression pistons of the dilatometry device, the stresses were corrected as [27]:(1)$$\sigma =\frac{\sigma_{0}C^{2}}{2\left[\exp\left(C\right)-C-1\right]}$$
where σ is the friction corrected true stress, and σ_0_ is the measured true stress. Parameter *C* can be determined as:(2)$$C=\frac{2mR}{h}$$

*R* and *h* are instantaneous radius and height of specimen, and *m* is the friction coefficient. Geometrical parameters can by determined according to the amount of barreling for specimen and the friction coefficient then is calculated as
(3)$$m=\frac{\left(\frac{R}{h}\right)b}{\left(\frac{4}{\sqrt{3}}\right)-\left(\frac{2b}{3\sqrt{3}}\right)}$$
where *R* is the theoretical radius, *h* is the final height of the sample after compression, and *b* is the barreling factor. The theoretical radius *R* can be determined as
(4)$$R=R_{0}\sqrt{\frac{h_{0}}{h}}$$
and barreling factor as
(5)$$b=4\frac{\Delta R}{R}\cdot \frac{h}{\Delta h}$$
where $R_{0}$ and $h_{0}$ are the initial dimension of sample, Δh is the difference between the initial and final height of the sample, and ΔR is the difference between the largest and smallest radii of the deformed sample.

The values of the geometric parameters of the sample before compression were $R_{0}$ = 5 mm and $h_{0}$ = 10 mm. After compression, the average values of the geometric parameters were ΔR = 0.81 mm and Δh = 5.93 mm. Based on these parameters, *m* = 0.358 and *C* = 0.564 were determined. Using these parameters and Equation (1), the values of the measured true stresses were corrected.

### 2.4. Constitutive Model

For this material, we chose a mathematical model based on the Arrhenius equation. This model is represented by the equation [28]
(6)$$Z=\dot{\varphi }\exp\left(\frac{Q}{RT}\right)$$
and
(7)$$\;\dot{\varphi }=\mathrm{AF}\left(\sigma \right)\exp\left(-\frac{Q}{\mathrm{RT}}\right)$$
where $\dot{\varphi }$ is strain rate, *A* is material constant, *Z* is Zener–Hollomon parameter, *R* is the gas constant (8.314 J/mol∙K), *Q* is the activation energy and acts as an indicator of deformation difficulty in the physical deformation theory, and *T* is the temperature [29]. The *Z* parameter represents the temperature compensated strain rate, which has been widely used to characterize the behavior of materials in hot working [30]. $F\left(\sigma \right)$ is a function of flow stress and can be described as follows:(8)$$F\left(\sigma \right)=\left\{\begin{array}{cc} \sigma^{n} & \alpha \sigma <0.8\\ \exp\left(\beta \sigma \right) & \alpha \sigma >1.2\\ {\left[\sinh\left(\alpha \sigma \right)\right]}^{n} & \mathrm{for}\;\mathrm{all}\;\sigma \end{array}\right.$$
where *n*, *β*, and *α* are material constants and can be determined from experimental flow stress data. The material constant *α* is determined using the parameters n’ and *β* defined as [31]
(9)$$n’={\left[\frac{\partial \ln\dot{\varphi }}{\partial \ln\sigma }\right]}_{T}$$
and
(10)$$\beta ={\left[\frac{\partial \ln\dot{\varphi }}{\partial \sigma }\right]}_{T}$$

Consequently, the parameter *α* is
(11)$$\alpha =\frac{\beta }{n’}$$

By combining Equations (7) and (8) for all stress levels (low and high), the strain rate can be expressed as
(12)$$\dot{\varphi }=A{\left[\sinh\left(\alpha \sigma \right)\right]}^{n}\exp\left(-\frac{Q}{RT}\right)$$
and its logarithm as
(13)$$\ln\left[\sinh\left(\alpha \sigma \right)\right]=\frac{\ln\dot{\varphi }}{n}+\frac{Q}{nRT}-\frac{\ln A}{n}$$

By differentiating Equation (13) at constant temperature,
(14)$$\frac{1}{n}=\left[\frac{\partial \ln\left[\sinh\left(\alpha \sigma \right)\right]}{\partial \ln\left(\dot{\varphi }\right)}\right]$$
where the parameter n can again be determined as the reciprocal value of the average slopes of the linear fitting between $ln[sinh\left(\alpha \sigma \right)$ and $ln\left(\dot{\varphi }\right)$. Moreover, from this dependence, parameter, A is determined from the intercept of the linear fitting according to
(15)$$\ln A=\frac{Q}{RT}+\frac{C´}{n}$$
where *C*’ represents the intercept of the linear fitting for each temperature. From Equation (12) it is possible to determine for each strain rate the dependence between $ln[sinh\left(\alpha \sigma \right)$ and 1000/*T*. Average value of the linear fitting slopes from this plot expresses the *Q*/*Rn*, and thus, the hot deformation activation energy *Q*. Then the flow stress based on the mathematical model will be
(16)$$\sigma =\frac{1}{\alpha }\ln\left\{{\left(\frac{Z}{A}\right)}^{\frac{1}{n}}+{\left[{\left(\frac{Z}{A}\right)}^{\frac{2}{n}}+1\right]}^{\frac{1}{2}}\right\}$$

## 3. Results and Discussion

The first results we focused on in our research were aimed at finding two initial temperatures: *Ac*_1_ and *Ac*_3_. The heating of all eight samples took place under the same conditions, i.e., the heating rate was set to 1 °C∙s^−1^. Figure 4a explains how we initially determined the temperatures of *Ac*_1_ and *Ac*_3_. The results showed that in the experimental steel AISI 4340, the start temperature of *Ac*_1_ was 725 °C, and the final temperature of austenite formation *Ac*_3_ was 763 °C.

The start and finish of the austenitization temperatures are shown to a greater extent in Figure 4b. The linear dilatation curves are crossed by tangent lines, which help us to accurately determine the initial temperatures when heating *Ac*_1_ and *Ac*_3_. Another helpful function is the derivation of the dilatation curve on which we can also observe the deviation of the dilatation curve from its linear direction. Using derivation of dilatation curves, we can more accurately determine the exact point of deviation of the curve.

After determining the initial temperatures, *A_c_*_1_ and *A_c_*_3_*,* on heating the samples, we focused on the following dilatation curves, which represented different cooling rates. These following eight dilatation curves investigate the decomposition of austenite at different cooling rates and the formation of the resulting microstructure. The formation of the final phases after cooling does not only have to be homogeneous but can be formed by a combination of several structures. The rate of cooling affects the formation of the final microstructure. Each resulting phase transformation has its own temperature ranges at which it can occur. These thermal ranges can represent a relatively wide range of formation of the follow structure, which occurs in the process of anisothermal decomposition of austenite.

From all eight curves, three dilatation curves for comparison were selected, which represent the limit values of the formation of individual phase transformations. At the highest cooling rate, i.e., 100 °C/s, determined martensite start temperature *Ms* was 321 °C. Figure 5a shows a green line of the dilatation curve, which is covered by a red tangent line. For a more accurate determination, the figure is supplemented by the superimposed derivation of the given dilatation curve (Figure 5a, black line). At this high cooling rate, the resulting structure was evaluated as martensitic with a small percentage of retained austenite, which was not further investigated in this paper.

Another evaluated curve was at a cooling rate of 0.5 °C∙s^−1^. We can observe that the initial temperature of martensite formation had a lower value, namely 292 °C (Figure 6a). With enough increase in the dilatation curve, a small dilatation change was also observed at 480 °C, which represents the beginning of the bainite transformation. According to the dilatation curve, the resulting microstructure is again martensitic with a very low bainite content (Figure 6b).

Lastly, we examined the dilatation curve at cooling of 0.01 °C∙s^−1^, which can be observed in Figure 7a. Up to three phase transformations occurred in this resulting structure, namely, ferritic, bainitic and martensitic. The ferritic began at 700 °C. Subsequently, a bainitic transformation began at temperature of 476 °C. The start of the martensitic transition did not occur until 158 °C. Final microstructure comprised bainite matrix with polygonal ferrite and a very small amount of martensite (Figure 7b).

The average temperature of *Ms* was determined from the dilatation curves in which only the martensite structure was formed. We used the first five cooling curves from 100 °C∙s^−1^ to 0.5 °C∙s^−1^. The results are shown in Figure 8. By averaging these five temperatures from the individual curves, the average temperature *Ms* was of 282 °C. We could not measure the temperature *Mf* on any curve, because the austenite to martensite transformation occurred below RT (room temperature). We can observe that as the cooling rate lowers, the temperature *Ms* also decreases.

### 3.1. CCT Diagram AISI 4340 Steel

The construction of the CCT diagram (Figure 9) of the experimental steel AISI 4340 was created from all eight expansion curves. For its accurate compilation, it was necessary to analyze dilatation curves and evaluate the microstructure of the measured samples, which confirms the formation of the phases in the dilatation curves. The austenitization temperature was set at 850 °C, and the heating rate of all samples was 1 °C∙s^−1^. To achieve the transformation of the austenite in the whole sample volume, the holding time at the austenitizing temperature was 30 min. We determined the temperatures of *Ac*_1_ and *Ac*_3_ using expansion curves during heating. The CCT diagram of experimental high-strength steel AISI 4340 consists of three transformation phases. The first phase occurred in the first four dilatation curves up to a cooling rate of 1 °C∙s^−1^ and consists only in martensite. We observed the formation of a second phase at a cooling rate of 0.5 °C∙s^−1^, which was a bainite. This resulting bainitic region continues until the last dilatation curve, which has been cooled at a rate of 0.01 °C∙s^−1^.

A third transformation phase also occurred in the CCT diagram, which is located above the bainitic region and is formed by a ferrite. The ferritic transformation region occurred only at the last three cooling curves, namely 0.1, 0.05 and 0.01 °C∙s^−1^ [32,33].

The end of all measurements was set at 50 °C. For this reason, in the CCT diagram, there is no boundary region of martensite formation *Mf*. This is because it lies in negative temperatures, and our device does not currently allow negative temperatures to be set. As already mentioned, the resulting *Ms* temperature was evaluated from the first dilatation curves and has a value of 282 °C. According to the CCT diagram, this temperature represents the limit value at which a martensite begins to form in the structure of the material. In order to avoid the formation of a possible heterogeneous structure, the specific cooling rate must be maintained.

We can also observe in the diagram that the hardness declined due to the decreasing cooling rate. This result is associated with the formation of the microstructure after cooling the samples. The highest Vickers hardness 850 was reached at the highest cooling rate of 100 °C∙s^−1^. The hardness results differed only slightly, where the microstructure was formed only by martensite. We observe that the hardness began to decrease after the beginning of the formation of a heterogeneous structure (bainite + martensite). Significant decrease in hardness occurred during the formation of ferrite. Final Vickers hardness of 423 was measured in the last dilatation curve, which was cooled at a rate of 0.01 °C∙s^−1^. This represents an almost 50% decrease compared to the highest cooling rate of 100 °C∙s^−1^. The arrangement of the ferrite atomic lattice is of the bcc type, while the martensitic structure creates a bct atomic lattice, which by its structure supports the increase of the material hardness [34,35,36,37].

Several authors have discussed the issue of creating a CCT diagram of a given steel. As is known, the main parameters that affect the overall shape of the CCT diagram are primarily the chemical composition. However, the set austenitization temperature is also an important factor. Boyer et al. [38] examined AISI 4340 steel from the point of view of creating its CCT diagram. The critical cooling rate at which only the martensite microstructure is formed was determined to be 8.3 °C∙s^−1^. Our critical rate of pure martensitic transformation is 0.5 °C. Overall, their diagram is shifted to the left. Likewise, a group of authors, Samadi Shahreza et al. [39] and Penha et al. [40], dealt with this issue, and their resulting CCT diagrams are also slightly shifted to the left. Nasar A. Ali [41] and Popescu et al. [42] investigate a very similar steel with a slightly higher content of Cr and Ni. Their CCT diagrams match our measured diagrams in the critical transformation of martensite as well as in the overall shape of the diagram. Our CCT diagram does not include the beginning of pearlitic transformation because this transformation takes place only at very low cooling rates. The higher content of Cr and Ni in the investigated steel shifts the whole area of pearlitic transformation to the right. When comparing our diagram, the value of the temperature *Ms* coincides with other authors. Our material contains the upper limits of the main alloying elements, and this higher percentage of alloying elements, especially Cr and Ni, significantly affects the resulting shape and orientation of our CCT diagram when compared to other authors.

### 3.2. True Stress–True Strain Diagrams of AISI 4340 Steels

Basic input parameters for high temperature True Stress–True Strain curves are temperature, deformation rate and deformation degree. Deformation degree was set as constant (Δ*L* = 0.6), while other two parameters were variable during the experiment. There are used temperatures 900, 1000, 1100 and 1200 °C and deformation rates 10, 1, 0.1, 0.01, 0.001 s^−1^ in the experiment. As a result, twenty True Strain–True Stress high temperature curves were measured. The measured flow stress data was corrected by Equation (1) due to the friction.

To create a constitutive model based on Arrhenius equation, it was necessary to determine the individual material constants. These constants are determined for each true strain value, in our case from *ϕ* = 0.1 to *ϕ* = 0.8 with a step of 0.05. The following will show the procedure for determining the parameters for true strain *ϕ =* 0.2. The parameter *n´* is obtained as the reciprocal value of the slope between ln(*σ*) and ln ($\dot{\varphi })$ (Figure 10a). The parameter *β* is again determined as reciprocal value of the slope between *σ* and ln ($\dot{\varphi })$ (Figure 10b). The average values of the slope determine the values of the parameters, namely *n´* = 5.603 and *β* = 0.0787. Then, according to Equation (11), *α* = 0.0141.

Based on Equation (14), a dependence was created (Figure 11a), from which the parameter *n* was determined as the average of the reciprocal values of linear fittings (*n =* 3.564). The activation energy value *Q* is determined based on the graphical dependence shown in Figure 11b. The average value of the activation energy for the material AISI 4340 was calculated *Q =* 314 kJ/mol. This value is lower compared to [43], where the value *Q* = 427 kJ/mol was obtained, where the values of flow stresses were not corrected by the effect of friction during compression. On the contrary, a lower value of activation energy, *Q* = 281.28 kJ/mol, was obtained by the authors [44], where, however, this value was determined by expression for determining the activation energy as a function of chemical composition developed by [45]. This calculation was created for general use for low-alloy steels and therefore does not provide as accurate results as the values of the activation energy obtained from the measured flow stresses.

The Zener–Hollomon parameter *Z* is determined according to Equation (6) for the respective true strain and the predicted flow stresses are calculated according to Equation (16) for all strain rate and deformation temperatures.

A comparison between the measured (lines) and experimentally obtained flow stresses (dots) is shown in Figure 12. The values and courses of stresses are highly dependent on the deformation temperature. A comparison of the curves shows that increasing strain rate or decreasing temperature causes an increase in flow stress. In other words, softening due to dynamic recrystallization and dynamic recovery is prevented and causes the material to exhibit strain hardening. The higher strain rate and lower temperature provide a shorter time for energy accumulation and lower mobility at the grain boundaries, which also leads to the growth of dynamically recrystallized grains and the partial removal of dislocations.

For each curve, after a rapid increase of stress to the highest (peak) value, it begins to gradually decrease towards a steady state with varying degrees of softening, indicating the onset of dynamic recrystallization (DRX). Furthermore, all flow stress curves can be divided into three stages. In the first stage, in which there is a sharp increase in stress up to a critical value, the work hardening dominates. In the second stage, the increase in stress is constantly slowed down to the peak value or the inflection value of the work hardening rate, which corresponds to the increasing influence of dynamic recovery and recrystallization. This effect increases and at some point exceeds the effect of work hardening. The highest peak stress was measured at the strain rate φ’ = 10 s^−1^ (328 MPa) and the lowest at the rate φ’ = 0.001 s^−1^ (65 MPa). This means that at the highest rate, the peak stress reached five times bigger value. In the last third stage, two predominant curves can be observed. One is a continuous decrease with a significant DRX mechanism, which occurs especially at the lowest temperature (900 °C) or higher transformation rates (1 s^−1^–10 s^−1^) in the entire range of deformation temperatures. The second is only a slight flow stress drop due to DRX. This course is especially visible at higher temperatures (1200–1100 °C). This is mainly because under these conditions, the higher speed of the DRX slows down the work hardening. This also shifts the peak and steady stresses to lower values of the true strain.

As can be seen from Figure 12, the predicted stress values using the Arrhenius-type constitutive model are in very good agreement with the measured values over the entire temperature range and strain rates. The Pearson’s correlation coefficient *R* was calculated to determine the quality of the correlation between the measured and predicted values of flow stresses. The evaluation of the suitability of using the selected model for flow stress prediction of AISI 4340 steel is shown in Figure 13. The value of the coefficient of determination in comparison with the best fitting is R^2^ = 0.9892.

Overall dependence of peak stress on temperature for individual strain rates is shown in Figure 14a. The peak stress values were obtained as the highest values of flow stresses from the dependences in Figure 12. The peak stress decreases linearly in relation to the deformation temperature. Similar results were obtained also by the authors [45]. Results clearly shows that the increasing temperature significantly affects deformation resistance of the material. When the material is heated to high temperatures in stable austenite area close to melting point, the interatomic distance increases. Distance between atoms in the crystal lattice increases, and consequently, strength of the material is reduced. Therefore, less force is required to deform the material. Figure 14b indicates the relationship between the peak stress and the logarithmic strain rate. The peak stress increases linearly in relation to the deformation rate. As with dependence on temperature in previous chapter, results clearly shows that the increasing deformation rate significantly affects deformation resistance of the material.

### 3.3. Microstructure Investigation

Microstructures of AISI 4340 for all deformation temperatures and two limiting values of deformation rates are in Figure 15 (*φ*’ = 10 s^−1^) and Figure 16 (*φ*’ = 0.001 s^−1^). The deformation temperature has a significant influence on the grain size [46]. In both shown cases of strain rates, grain size increases in proportion to temperature. This effect is more noticeable as the deformation rate decreases.

Microstructures of all deformation rates and two limiting values of temperatures are in Figure 17 (*T* = 900 °C) and Figure 18 (*T* = 1200 °C). The deformation rate has a lower effect on grain size than temperature. This effect can be observed in (Figure 16a–c) where the individual samples differ only minimally due to the changing deformation rate.

The dynamic recrystallization occurs at higher temperatures, which suppresses the effect of deformation on the microstructure. Therefore, the microstructures are coarser at higher temperatures regardless of the deformation rate.

The combination of highest deformation rate *φ*´ = 10 s^−1^ and lowest temperature *T* = 900 °C has the most significant observed effect on grain size. At this temperature, there is no dynamic recrystallization present, and the grains remain deformed.

The correlations hardness versus logarithm of the strain rate (Figure 19a) and hardness versus deformation temperature (Figure 19b) indicate to be linear. As can be seen, the hardness increases linearly as the logarithm of strain rate rise. Diagram of final hardness versus peak stress σ_p_ (Figure 20) showed linear correlation between those variables.

## 4. Conclusions

The behavior of AISI 4340 steel under various temperature and deformation conditions was investigated in this paper. Dilatometry analysis was designed to construct the CCT diagram of a high-strength steel from eight dilatation curves. The cooling rate was set from 100 °C/s to 0.01 °C/s. Further analysis examined hot deformation behavior at various deformation temperatures from 900 ° C to 1200 °C and strain rates from 0.001 s^−1^ to 10 s^−1^. The following conclusions can be drawn from the present work:Transformation temperatures have values of *Ac*_1_ = 725 °C to *Ac*_3_ = 763 °C. The selected austenitizing temperature thus guarantees full austenitizing throughout whole material bulk since it was set at 850 °C and holding time at this temperature was 30 min.Average martensitic transformation start temperature is *Ms* = 282 °C. The temperature is calculated from five dilatation curves (100, 10, 5, 1 and 0.5 °C/s) with martensitic transformation presence only. The *Ms* temperature softly decreases with reduction of cooling rate.The highest Vickers hardness value of 850 was reached at the highest cooling rate of 100 °C/s and, conversely, at the lowest cooling rate of 0.01 °C/s the Vickers hardness of approximately half was reached, namely 423. The hardness decreases with decreasing of cooling rate, and this result is associated with the formation of individual structures.Deformation rate has also significant effect on peak stress, which increases exponentially with increase in deformation rate. The highest value of peak stress (*σ_p_* = 328 MPa) was measured with a combination of the highest strain rate (*φ*´ = 10 s^−1^) and the lowest temperature (*T* = 900 °C). The Arrhenius-type constitutive model can be used for prediction flow stress values of this type of steel with high prediction accuracy.Microstructure investigations show that dynamic recrystallization takes a place at higher temperature (1200 °C), and grain coarsening occurs regardless of strain rate. Furthermore, high strain rate causes grain refinement. This effect is most noticeable at lower temperatures (900 °C) where the recrystallization is not present, and grains remain deformed with the result of very fine grains even after transformation of austenite to martensite or other phases.The most noticeable effect on microstructure was observed with combination of the highest strain rate *φ*´ = 10 and the lowest temperature *T* = 900 °C. This combination of deformation rate and temperature shows the most grain-refined microstructure.The basic conclusion of the deformation analysis is that the peak stress decreases linearly with increasing deformation temperature and increases linearly with the deformation rate. The peak stress σ_p_ is very closely related to Vickers hardness where their dependence is linear.

## Figures and Tables

**Figure 1 materials-13-05585-f001:**
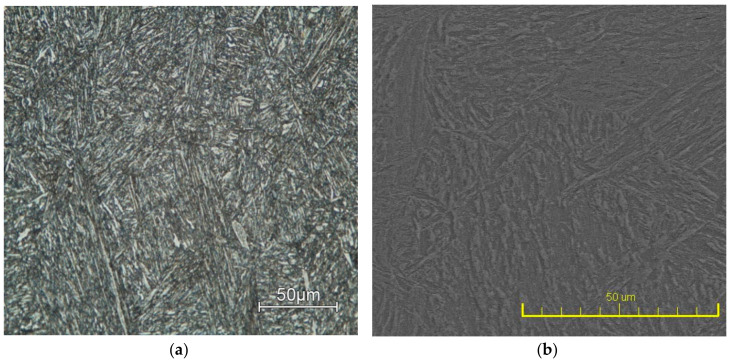
Microstructure of base material AISI 4340, etching by 3% Nital: (**a**) LOM; (**b**) SEM.

**Figure 2 materials-13-05585-f002:**
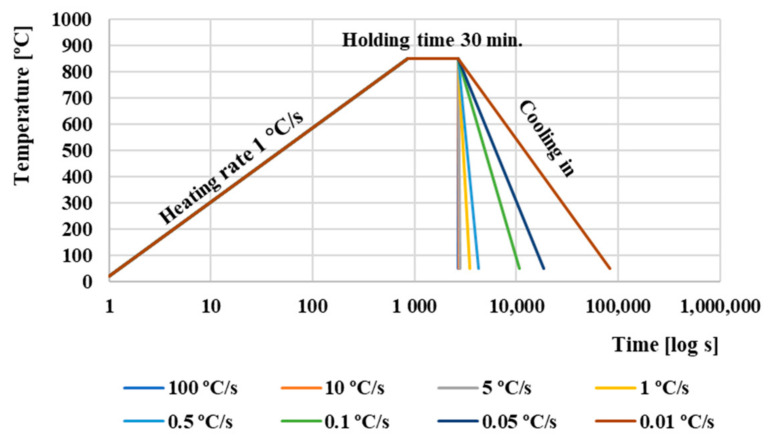
Heating and cooling regimens used for construction the CCT diagram of AISI 4340 steel.

**Figure 3 materials-13-05585-f003:**
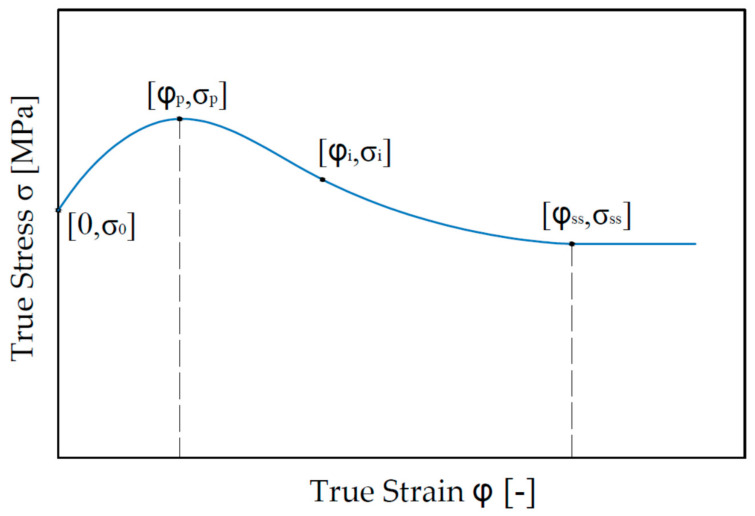
Typical True stress–True strain dependence of steels.

**Figure 4 materials-13-05585-f004:**
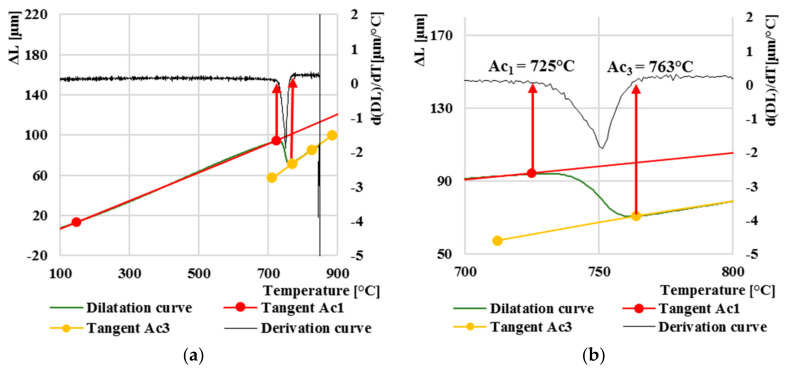
(**a**) Dilatation curve on heating; (**b**) Enlarged part (zooming) of the length change vs. temperature curve–determination of *Ac*_1_ and *Ac*_3_ temperatures.

**Figure 5 materials-13-05585-f005:**
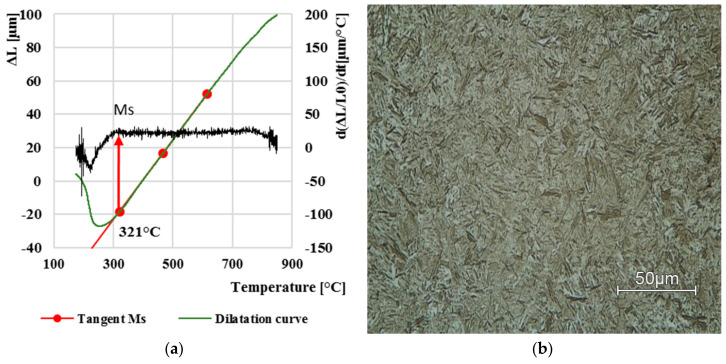
Dilatation curve (**a**); the final microstructure LOM (**b**) after cooling at 100 °C·s^−1^.

**Figure 6 materials-13-05585-f006:**
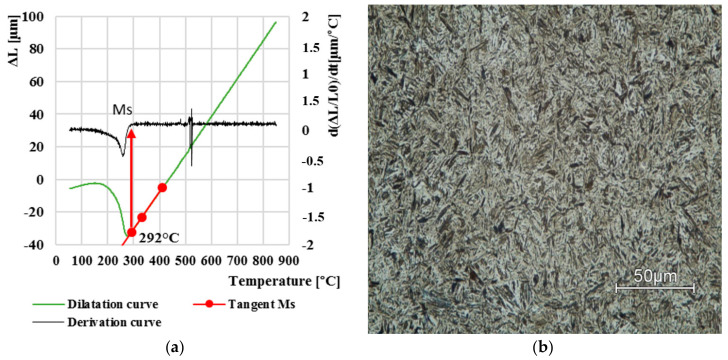
Dilatation curve (**a**); the final microstructure LOM (**b**) after cooling at 0.5 °C·s^−1^.

**Figure 7 materials-13-05585-f007:**
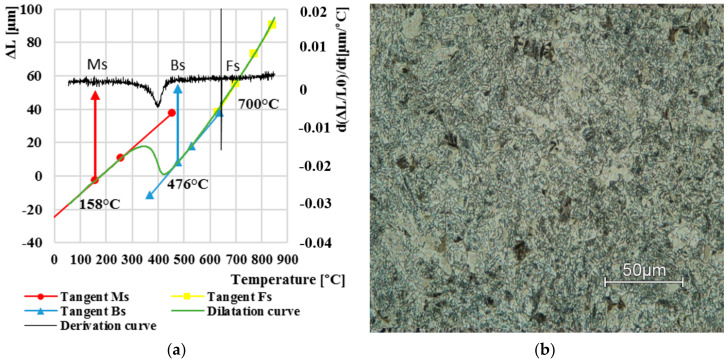
Dilatation curve (**a**); the final microstructure LOM (**b**) after cooling at 0.01 °C·s^−1^.

**Figure 8 materials-13-05585-f008:**
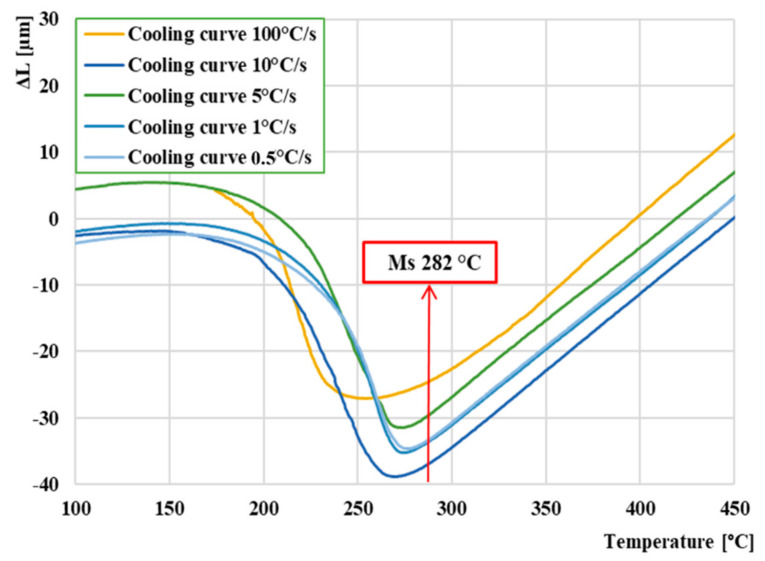
Comparison and determination of average temperature *Ms.*

**Figure 9 materials-13-05585-f009:**
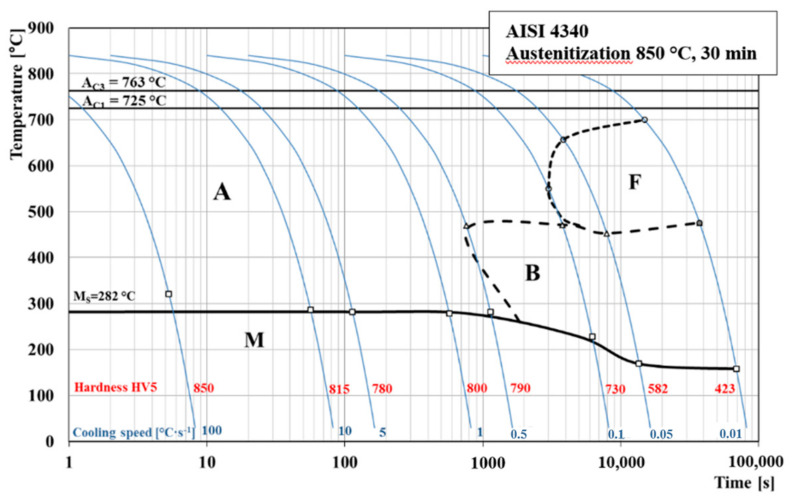
CCT diagram of AISI 4340 steel.

**Figure 10 materials-13-05585-f010:**
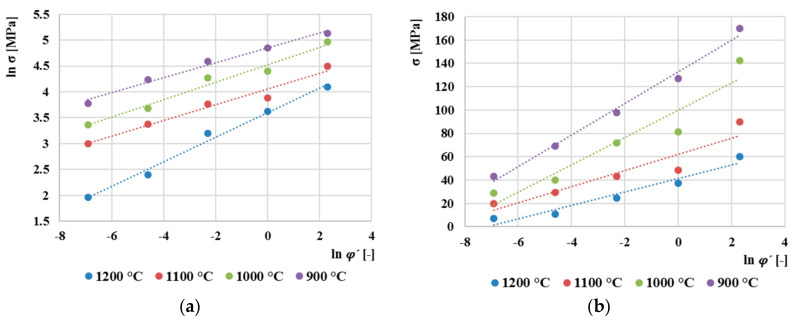
(**a**) Dependence between ln $\dot{\varphi }$ and ln*σ* for derivation of *n´* parameter, (**b**) dependence between ln $\dot{\varphi }$ and *σ* for derivation of *β* parameter.

**Figure 11 materials-13-05585-f011:**
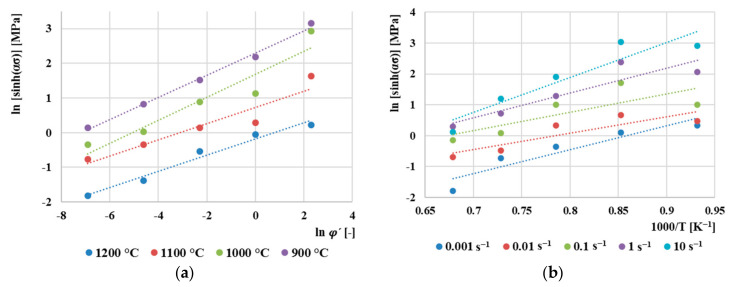
(**a**) Dependence between $\ln[\sinh\;\left(\alpha \sigma \right)$ and $\ln\left(\dot{\varphi }\right)$ for derivation of *n* parameter, (**b**) dependence between $\ln[\sinh\;\left(\alpha \sigma \right)$ and 1000/*T* for derivation *Q* parameter.

**Figure 12 materials-13-05585-f012:**
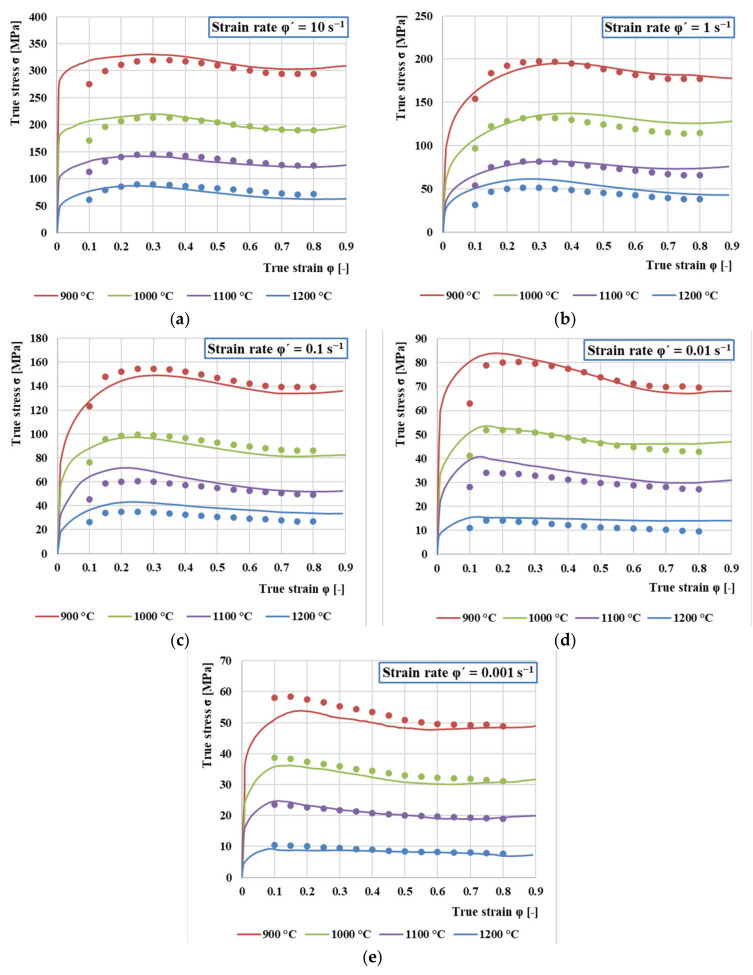
True Stress–True Strain Diagrams with measured (lines) and predicted (dots) flow stresses for (**a**) strain rate *φ*’ = 10 s^−1^; (**b**) strain rate *φ*’ = 1 s^−1^; (**c**) *φ*’ = 0.1 s^−1^; (**d**) *φ*’ = 0.01 s^−1^; (**e**) *φ*’ = 0.001 s^−1^.

**Figure 13 materials-13-05585-f013:**
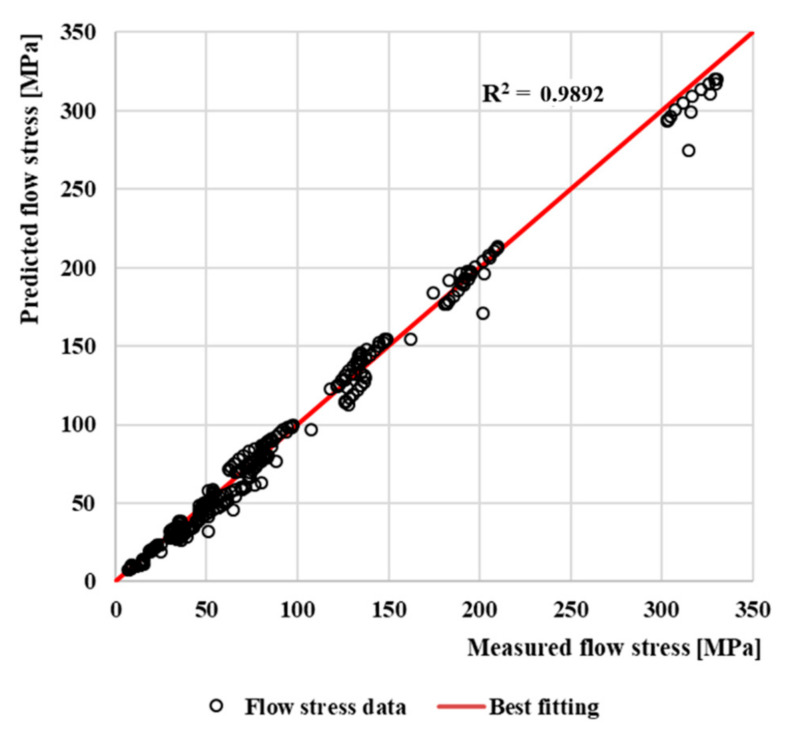
The correlation between predicted and measured flow stresses and comparation with best fitting.

**Figure 14 materials-13-05585-f014:**
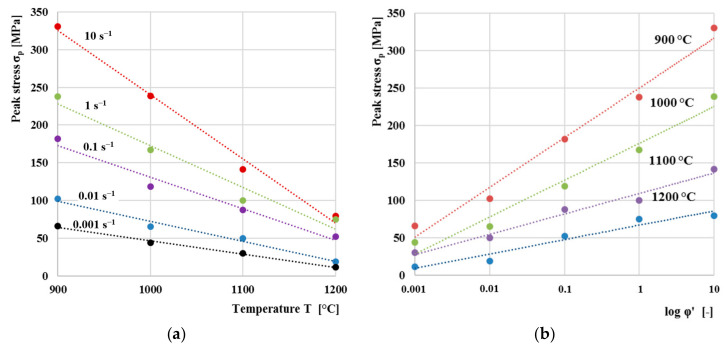
(**a**) dependence of strain resistance on temperature for individual strain rates; (**b**) dependence of strain resistance on strain rates for individual temperatures.

**Figure 15 materials-13-05585-f015:**
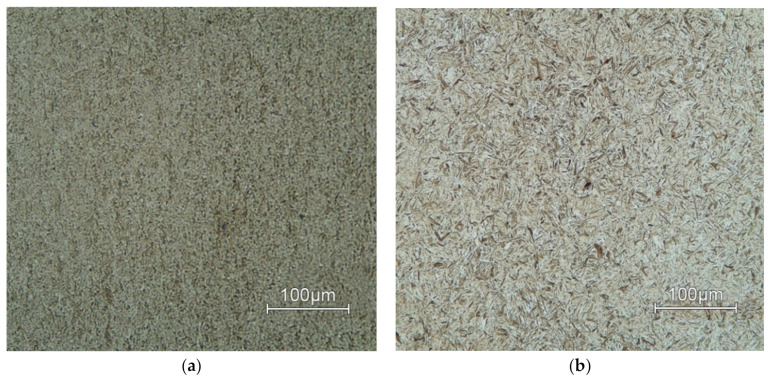
Microstructure of samples deformed at strain rate *φ*’ = 10 s^−1^ at temperature: (**a**) *T* = 900 °C; (**b**) *T* = 1000 °C; (**c**) *T* = 1100 °C; (**d**) *T* = 1200 °C.

**Figure 16 materials-13-05585-f016:**
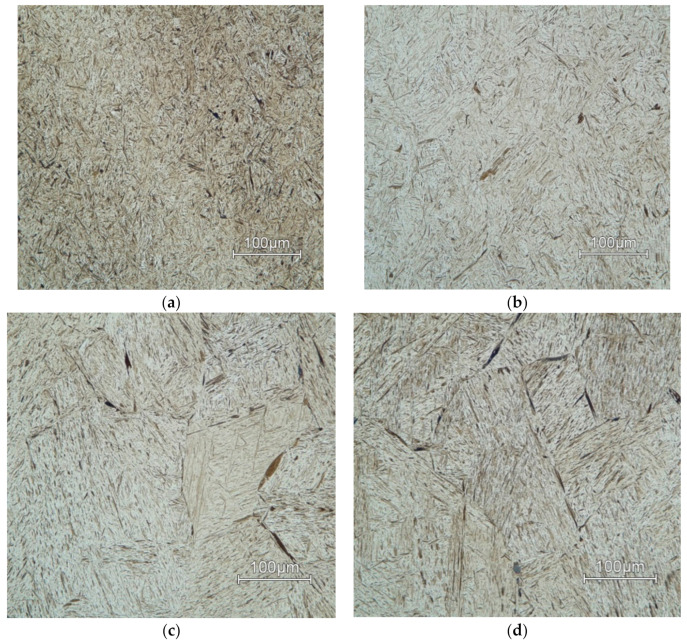
Microstructure of samples deformed at strain rate *φ*´ = 0.001 s^−1^ at temperature of: (**a**) *T* = 900 °C; (**b**) *T* = 1000 °C; (**c**) *T* = 1100 °C; (**d**) *T* = 1200 °C.

**Figure 17 materials-13-05585-f017:**
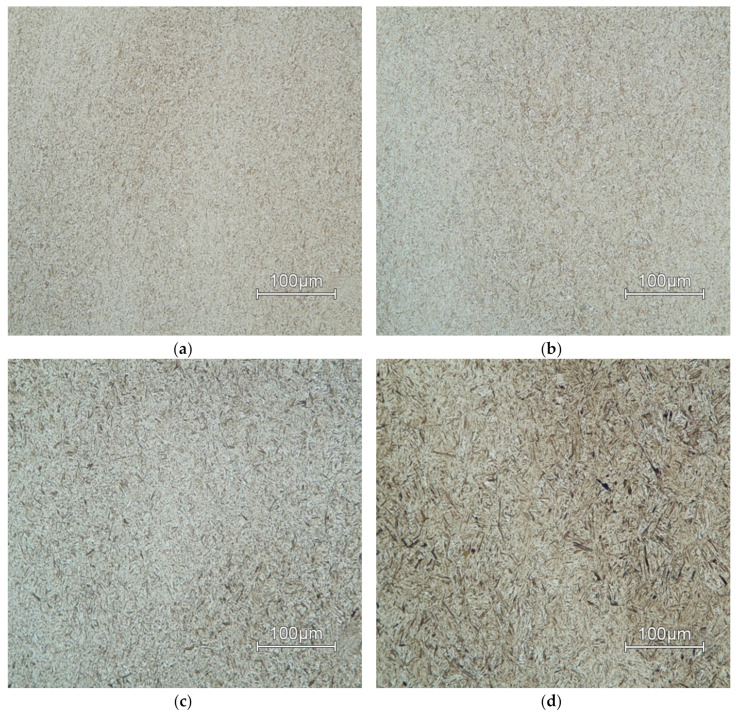
Microstructure of samples deformed at temperature of *T* = 900 °C at strain rate: (**a**) 1 s^−1^; (**b**) 0.1 s^−1^; (**c**) 0.01 s^−1^; (**d**) 0.001 s^−1^.

**Figure 18 materials-13-05585-f018:**
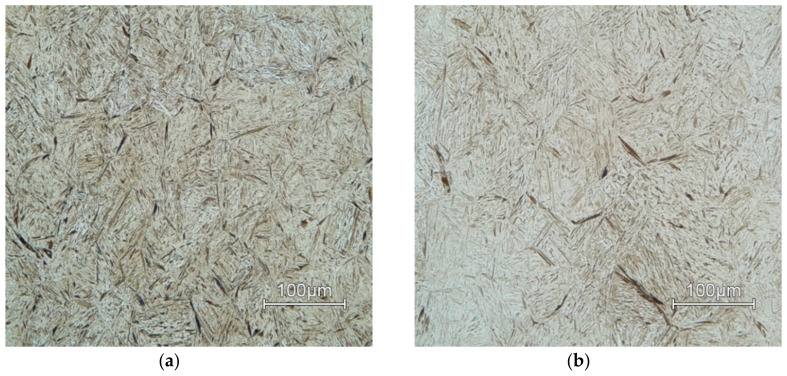
Microstructure of samples deformed at temperature of *T* = 1200 °C at strain rate: (**a**) 1 s^−1^; (**b**) 0.1 s^−1^; (**c**) 0.01 s^−1^; (**d**) 0.001 s^−1^.

**Figure 19 materials-13-05585-f019:**
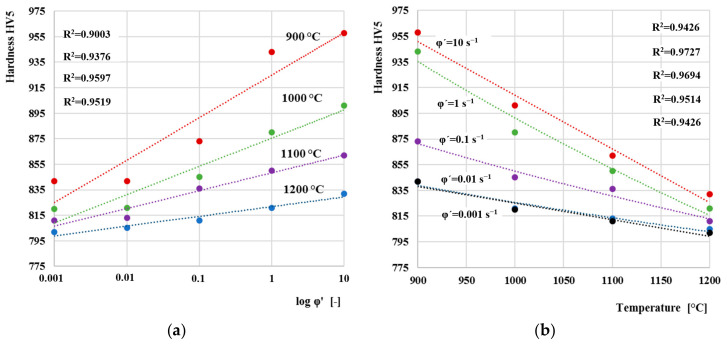
Influence of strain rate (**a**) and deformation temperature (**b**) on Vickers hardness (HV5) of AISI 4340 steel.

**Figure 20 materials-13-05585-f020:**
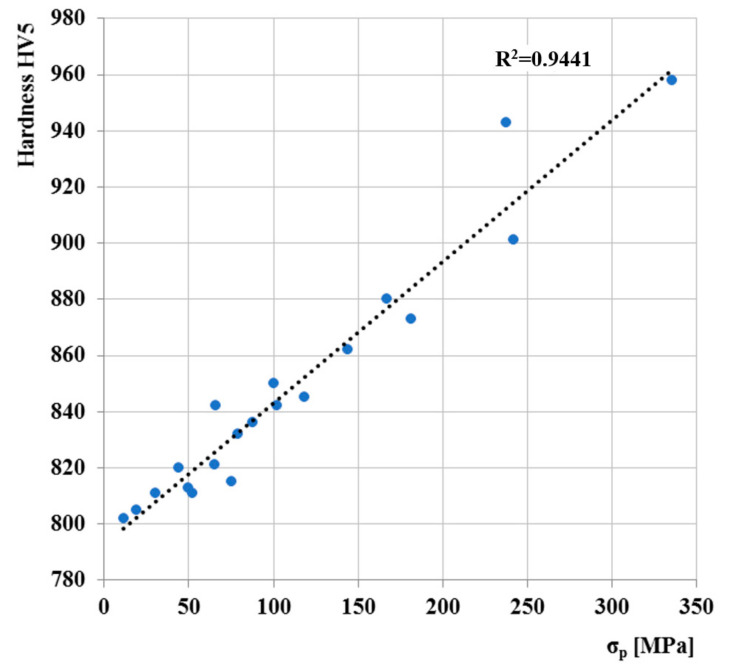
Total dependence of peak stress σ_p_ on the Vickers hardness.

**Table 1 materials-13-05585-t001:** Chemical composition of the AISI 4340 examined steel (wt.%).

Element	C	Mn	Si	Cr	Ni	Mo	V
Min	0.33	0.25	0.17	1.20	1.65	0.35	0.10
Max	0.40	0.50	0.37	1.50	2.00	0.45	0.80
Spectral analysis	0.40	0.30	0.32	1.49	1.98	0.52	0.13

**Table 2 materials-13-05585-t002:** Basic properties of the AISI 4340 steel.

Tensile Strength *R_m_* (MPa)	Modulus of Elasticity *E* (GPa)	Thermal Conductivity (W·m^−1^·K^−1^)	Hardness (HV5)	Specific Heat (J·kg^−1^·K^−1^)
1500	210	20	596	460

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
