# Peer review of "Effect of Selected Cooling and Deformation Parameters on the Structure and Properties of AISI 4340 Steel"

_materials, 2020, doi:10.3390/ma13235585_

Round 1

Reviewer 1 Report

The paper presents the investigation into the phase transformation temperatures and the deformation parameters of the AISI 4310 steel. The research material and the idea of the paper are interesting, however, in my opinion, the paper is written like a technical report. The "scientific" idea should be much emphasized. A more scientific paper should be used for discussion and introduction. The novelty of the paper should be underlined. Conclusions must be improved too. 

Comments on the paper.

  1. Title: instead of "some" use "selected"
  2. The introduction is interesting but contains basic information. The introduction contains basic knowledge and does not justify the goal of the work. It should be improved by adding some more scientific results, discussion etc. What are the drawbacks in the literature / scientific papers?. What is the state of art? Please add it.
  3. The introduction does not present / justify the "scientific goal" of the work. The aim of the work is poorly justified. What is the state of art in the forming, dilatometry research in the Cr-Mo-'M' steels? 
  4. In current form, the scientific-discussion is missing in the paper (in the Results section). The paper does not contain the discussion of the results with the other papers. The discussion of the obtained results should be added. Without it, the paper looks rather like a technical report. Please improve it.
  5. The following papers can be useful for improving the introduction and discussion:
    1. https://doi.org/10.3390/met8040232
    2. https://doi.org/10.3390/ma13092022
    3. 10.13140/RG.2.2.20464.25607
  6. The elaboration of CCT diagram of the steel has little scientific value. There are many works describing this diagram. On the other hand, you can improve your paper by discussing your CTT diagram shape with the literature references. It should be done.
  7. The temperature of austenitization is not given in the text. The volume of the billet plus time of austenitization can affect the microstructure. Add the temperature and holding time. Please consider adding the time-temperature plot of heat treatment.
  8. Conclusions are too broad. Please select the most important ones and shorten to make it much more concise. Also, see my comments on a few conclusions: 
    1. Eg. conclusion 1 is unclear. What were the time and the temperature of austenitization - it is not given in the text. 
    2. Eg. Conclusion 3 is trivial. Should be deleted.
    3. eg. Conclusion 4 is too general and "generally" obvious. Should be precise. 

Author Response

Dear Sir or Madam,

thank you for your useful comments and constructive criticism on our manuscript Effect of selected cooling and deformation parameters on the structure and properties of AISI 4340 steel”. Your recommendations will definitely further improve the overall quality of our manuscript.

Point 1: The manuscript title was changed.

Point 2 and Point 3: The introduction was supplemented by literature and the contribution of the article.  

Point 4: Scientific discussion was added in the Results section.

Point 6: Discussion of the CCT diagram shape with literature references was added.

Point 7: Austenitization temperature was added to the manuscript as well as time-temperature plot of heat treatment.

Point 8: The conclusion was shortened, but due to the number of experiments performed within the article, the number of conclusions is also larger. It has also been modified based on your comments.       

Thank you very much.

Yours sincerely,

Ing. Maroš Eckert, PhD.

Faculty of Special Technology

Alexnader Dubcek University of Trencin 

Reviewer 2 Report

The work contains interesting and important results, first, the construction of CCT diagram for AISI 4340. The phase transitions during cooling at different rates also represent big interest. Other results probably are also interesting and important but they are not sufficiently discussed.

Author must pay attention on that:

Equations 1-3 all are taken form ref [21], but is it necessary to mention thta 3 times?

Why eq.1 - 3 are presented in manuscript if they are not used in calculations?

English: Line 168 “…material constrants…”  Line 227 „...and also the evaluate the microstructure…”

Where from eq.4-6 are taken? Reference is not given. That is scientific meaning of them. What represents eq.(4)? What is fi? What is Z in eq.6, how it is obtained. Coefficients in eq.4,5 are not explained. They are not explained also in ref. [22] as indicated. In fact, nothing ae written about eq.4-6 in ref [22], it is wrong referencing. It is necessary to explain physical meaning of all coefficients of eq.4-6 and indicate the source of them, to give the true reference.

In Figure 9 many curves are presented but with very short comments. What reader must learn from those curves?

In Figure 9 there are calculated curves obtained according to eq.(6). They can be interesting, but calculation parameters and its values are not indicated and not analyzed. There are many curves, but which parameters were changed? What the physical meaning is of that change of parameters what we learn from that. Datail explanation on that is necessary. Calculation details must be added and explained. The discussion must be extended on those results, how they fits experiment, why deviates. The same with figure 10.

Are results in Figure 11 just selected data from figures 9 and 10? If yes, please clearly indicate method of data selection, which points are summarized.

Line 306, there is indication to not existing figure 20b

Many pictures but very little discussion about figures 13, 14, 15, 16. Microstructures from light microscope are presented but conclusions are done about grain size which is not very well seen form those pictures. Why the direct quantitative measurements of grain size were not performed? There are many good and simple methods for that, e.g. XRD. Even form presented light microscope pictures the grain size it is possible to show for readers visually and estimate quantitatively. Authors could do more attempts on that.

Authors must explain the details how figure 17 using data from figures 13-16 was drawn.

More detail discussion on figures 17, 18 is needed.

Conclusion 3: How it can be proven that in Final CCT diagram of high strength AISI 4340 steel shows three different phases, namely 343 martensitic, bainitic and ferritic. Form the presented measurement it is seen only fact that phase transition occurs but which phase is formed is not obvious. The measurements to show the type of phases were not performed. Authors must add more discussion on that.

Author Response

Dear Sir or Madam,

thank you for your useful comments and constructive criticism on our manuscript Effect of selected cooling and deformation parameters on the structure and properties of AISI 4340 steel”. Your recommendations will definitely further improve the overall quality of our manuscript.

We have tried to incorporate your observations and comments into our manuscript. We analyzed in more detail and explained the individual parameters and the method of obtaining material parameters into the constitutive model. We also described in more detail the flow stress curves, as well as CCT diagrams, and the obtaining of peak stresses. The data needed to create the dependence between the hardness and temperature resp. strain rate are, in our opinion, sufficiently described in the section materials and methods. These dependencies are to serve mainly as input parameters of simulations and modelling of technological processes by other authors. We also adjusted the conclusion according to your comments.

We hope that we have revised the paper to satisfy all your comments and suggestions and therefore we would kindly like to ask you for accepting our revised version and considering it for publication.

Thank you very much.

Yours sincerely,

Ing. Maroš Eckert, PhD.

Faculty of Special Technology

Alexnader Dubcek University of Trencin 

Reviewer 3 Report

Authors did a reliable and complete work. However, the scale bars are not very good. Please redo the scale bars in all pictures. English should be proofread.

Author Response

Dear Sir or Madam,

thank you for your useful comments and constructive criticism on our manuscript Effect of selected cooling and deformation parameters on the structure and properties of AISI 4340 steel”. Your recommendations will definitely further improve the overall quality of our manuscript.

Point 1: All scale bars have been adjusted.

Point 2: English has been corrected.

We hope that we have revised the paper to satisfy all your comments and suggestions and therefore we would kindly like to ask you for accepting our revised version and considering it for publication.

Thank you very much.

Yours sincerely,

Ing. Maroš Eckert, PhD.

Faculty of Special Technology

Alexnader Dubcek University of Trencin 

Round 2

Reviewer 1 Report

Dear Authors,

Generally, I accept your explanations. However, your responses are too scant. You should provide a full explanation to review comments - it will save the reviewer time and facilitate the re-reviewing process. Here are my comments on the paper.

  1. I think that your austenitization time/temperature parameters 0.5h/850C results in significant grains growth. Don't you think that the time was longer than required for such a small sample? According to the heat treatment procedures, only 2min per 1 mm of sample is recommended. In your case 8 minutes guarantee austenitization without grains coarsening. What do you think about it? Can it affect the CCT diagram? Looking for your response.
  2. It is a pity that you don't measure the grain size in various processes. Can you compare it quantitatively? Explain it.
  3. Conclusion#1: The size of the sample as well as holding time are crucial for austenitization. Please add the time of the austenitization.
  4. Conclusion#3: delete it - it is quite a general statement. 
  5. Conclusion#4: Add the phrase that "Hardness decreases with
    437 decreasing cooling rate"
  6. In the discussion (fig20) and results sections, please change the "hardness HV5" to "Vickers hardness" or simply "hardness". The reader is able to check the specific hardness load in the "materials and method section"
  7. This phrase should be precisely stated "where the recrystallization is not present, and grains remain deformed with the result
    452 of very fine grain size even after transformation of austenite to martensite or other phases". What it means "the fine grain size" - provide the value of eg. ferret diameter. Wat means "grain coarsening"? can you add the size of grain?

Author Response

Dear Sir or Madam,

thank you for your useful comments and constructive criticism on our manuscript Effect of selected cooling and deformation parameters on the structure and properties of AISI 4340 steel”. Your recommendations will definitely further improve the overall quality of our manuscript.

Point 1: The austenitization temperature and the holding time at this temperature were chosen in order to approximate the real conditions of heating and cooling of the components from practice, which are the subject of the research. In industry and practice, large parts are used where the holding time varies even in hours. No significant grain coarsening was noted when evaluating the samples. The steel contains carbides (it is alloyed with carbide-forming Cr), which inhibit grain growth. The heating conditions were constant for all monitored cooling processes, therefore the possible coarsening of the grain has no effect in terms of their mutual comparison. The holding time at the austenitization temperature, in contrast to the height of the austenitization temperature and the chemical composition of the material, does not have a significant effect on the final shape of the CCT diagram, which is confirmed by the work of other authors.

Point 2:  We chose an indicative comparison of grain size by the visual method from images of microstructures taken under the same conditions and with the same scale. In some cases, the structure is very complex and it is difficult to detect the boundaries of primary austenitic grains. In selected cooling processes, it would be possible to quantify the grain size by the size of the martensitic needles, but this quantification would not correspond to the cooling modes in the presence of bainite and ferrite.

Point 3: Modified according your comment.

Point 4: Modified according your comment.

Point 5: Modified according your comment.

Point 6: Modified according your comment.

Point 7: We modified it to “fine grains”. It is the same case that we did not quantitatively evaluate the grain size, only using a visual method with the same scale of the images. Also, achieving PAG visibility in such a wide range of structures requires several different approaches to grain boundary detection. Therefore, in the manuscript we used only the terms fine and coarse grains to compare them. But we will certainly use the available methods to accurately quantify grain size in the future.

Thank you very much.

Yours sincerely,

Ing. Maroš Eckert, PhD.

Faculty of Special Technology

Alexnader Dubcek University of Trencin 

Reviewer 2 Report

the manuscript has been significantly
improved and now warrants publication in Materials

Author Response

Dear Sir or Madam,

thank you for your useful comments and constructive criticism on our manuscript Effect of selected cooling and deformation parameters on the structure and properties of AISI 4340 steel”. Your recommendations improved the overall quality of our manuscript.

Thank you very much.

Yours sincerely,

Ing. Maroš Eckert, PhD.

Faculty of Special Technology

Alexnader Dubcek University of Trencin